# Differential Alternative Splicing Genes in Response to Boron Deficiency in *Brassica napus*

**DOI:** 10.3390/genes10030224

**Published:** 2019-03-18

**Authors:** Jin Gu, Wei Li, Sheliang Wang, Xiaoyan Zhang, Anne Coules, Guangda Ding, Fangsen Xu, Jian Ren, Chungui Lu, Lei Shi

**Affiliations:** 1National Key Laboratory of Crop Genetic Improvement, Huazhong Agricultural University, Wuhan 430070, China; gujin@yearth.cn (J.G.); henry_xiaowei@sina.com (W.L.); sheliangwang2017@mail.hzau.edu.cn (S.W.); dgd@mail.hzau.edu.cn (G.D.); fangsenxu@mail.hzau.edu.cn (F.X.); 2Microelement Research Center/Key Laboratory of Arable Land Conservation (Middle and Lower Reaches of Yangtze River), Ministry of Agriculture, Huazhong Agricultural University, Wuhan 430070, China; 3School of Animal, Rural and Environmental Sciences, Nottingham Trent University, Nottingham NG25 0QF, UK; xiaoyan.zhang2017@my.ntu.ac.uk (X.Z.); anne.coules@ntu.ac.uk (A.C.); 4State Key Laboratory of Biocontrol, School of Life Sciences, Sun Yat-sen University, Guangzhou, Guangdong 510275, China; renjian@sysucc.org.cn

**Keywords:** boron deficiency, alternative splicing, differential expressed genes, differential alternative splicing genes, splicing factors, *Brassica napus*

## Abstract

Alternative splicing (AS) can increase transcriptome diversity, protein diversity and protein yield, and is an important mechanism to regulate plant responses to stress. Oilseed rape (*Brassica napus* L.), one of the main oil crops in China, shows higher sensitivity to boron (B) deficiency than other species. Here, we demonstrated AS changes that largely increased the diversity of the mRNA expressed in response to B deficiency in *B. napus*. Each gene had two or more transcripts on average. A total of 33.3% genes in both Qingyou10 (QY10, B-efficient cultivar) and Westar10 (W10, B-inefficient cultivar) showed AS in both B conditions. The types of AS events were mainly intron retention, 3′ alternative splice site, 5′ alternative splice site and exon skipping. The tolerance ability of QY10 was higher than that of W10, possibly because there were far more differential alternative splicing (DAS) genes identified in QY10 at low B conditions than in W10. The number of genes with both DAS and differentially expressed (DE) was far lower than that of the genes that were either with DAS or DE in QY10 and W10, suggesting that the DAS and DE genes were independent. Four Serine/Arginine-rich (SR) splicing factors, *BnaC06g14780D*, *BnaA01g14750D*, *BnaA06g15930D* and *BnaC01g41640D*, underwent differentially alternative splicing in both cultivars. There existed gene–gene interactions between *BnaC06g14780D* and the genes associated with the function of B in oilseed rape at low B supply. This suggests that oilseed rape could regulate the alterative pre-mRNA splicing of SR protein related genes to increase the plant tolerance to B deficiency.

## 1. Introduction

Alternative splicing (AS) is a widespread phenomenon in the majority of eukaryotic organisms that can generate one or multiple mRNAs isoforms from the same precursor mRNA (pre-mRNA) by using different splice sites [1,2]. The basic splicing process includes assembly of spliceosome and splicing [3]. Splice sites recognition of constitutive splicing by spliceosomes is regulated by 5′ and 3′ consensus cis-sequences and branchpoint [1]. Nevertheless, alternative splicing splice-site selection is also determined by cis-acting elements and trans-acting factors, which contain exonic/intronic splicing enhancers (ESEs/ISEs) and exonic/intronic splicing silencers (ESSs/ISSs), respectively. These sequences are recognized by Serine/Arginine-rich (SR) proteins and heterogeneous nuclear ribonucleoproteins (hnRNPs) that are associated with trans-acting factors and negative-acting factors. These trans-acting and negative-acting factors can promote or inhibit intron removal [4,5]. The different percent of regulatory cis-elements and differential expression of splicing factors in different cell types, tissues, developmental stages, and environmental conditions affect AS patterns that significantly increase transcriptome and proteome diversity [6,7].

With the development of sequencing technique, more and more AS genes have been detected in plants. Four major types of AS events have been identified in plants: intron retention, alternative 3′ splice site, alternative 5′ splice site and exon skipping. Exon skipping is seen in the vast majority of the AS events [8]. However, intron retention is the most common form [9,10,11]. In *Arabidopsis*, 42–61% of the intron-containing genes are alternatively spliced [10,12]. Most AS isoforms that have a premature termination codon (PTC) are degraded by the nonsense-mediated mRNA decay (NMD) surveillance machinery, which can regulate the functional transcripts expression through a process termed regulated unproductive splicing and translation (RUST) [13,14]. It is reported that 13–18% of the introns of genes are alternatively spliced and regulated by NMD pathway in *Arabidopsis*, such as *CIRCADIAN CLOCK-ASSOCIATED1* (*CCA1*) that is a MYB-related transcription factor [12].

Plant stress-related genes are particularly prone to AS events, which often modulate the ratio between active and inactive isoforms in response to abiotic stress, thus fine-tuning the expression of key stress regulators [15]. Many alternatively spliced isoforms of AS genes show a tissue-specific expression and some of them are differentially expressed (DE) in plants in abiotic stress [7,16]. Recent genetic and transcriptomic analyses have identified important roles for numerous splicing factors in the control of plant abiotic stress responses [7,11,15].

Boron (B) deficiency is a major problem in China and worldwide, which causes the low yield and poor quality of many crops [17]. In vascular plants and diatoms, B plays a key role in the physical and biochemical process, such as cell wall synthesis, cell membrane integrity, carbohydrate metabolism, phenols metabolism, pollen germination and pollen tube growth [18,19]. Oilseed rape (*Brassica napus* L.) requires much more B than other species for growth and development and is sensitive to B deficiency [20]. A B-efficient *B. napus* cultivar Qingyou10 (QY10) and a B-inefficient *B. napus* cultivar Westar10 (W10) were screened by two-pace method (biomass screens at seedling and seed yield assessment at maturity) under contrasting B supply [21]. Proteomic analysis in the roots of QY10 with B deficiency in both long-term and short-term revealed the proteins in response to B deficiency involved in transporter, cell wall structure, defense related protein, carbohydrate and energy metabolism, antioxidant and detoxification mechanism, signaling and regulation, amino and nucleic acids metabolism and fatty acid metabolism [22,23]. qBEC-A3a, a major QTL for B efficiency, plays a significant role in improving B efficiency under B deficient conditions [24,25]. Exploiting quantitative trait locus (QTL) fine mapping and digital gene expression (DGE) analyses identified a nodulin 26-like intrinsic protein gene (*NIP*), which encodes a likely boric acid channel. A co-expression network analysis of the putative B transporters also highlighted its central role in the B uptake efficiency of *B. napus* [25]. Moreover, the B transporter BnaC4.BOR1;1c is critical for inflorescence development and fertility under B limited conditions in *B. napus* [26].

Alternative splicing is an important mechanism to regulate plants responses to abiotic stress [7,15,27,28,29]. The AS genes of *B. napus* under B deficiency have not been reported. In this study, RNA-Seq was used to investigate the AS events and DAS events in response to B deficiency in *B. napus* cultivars QY10 and W10. This should contribute to our understanding of the B deficiency response mechanism in oilseed rape, and therefore assist in efforts to improve B-efficiency in *B. napus*.

## 2. Materials and Methods 

### 2.1. Plant Materials and Growth Conditions

B-efficient *B. napus* cultivar Qingyou 10 (QY10) and B-inefficient *B. napus* cultivar Westar 10 (W10) were used in this study. The seedlings were cultured using Hoagland solution [30] in a growth chamber. Then, 25 µL H_3_BO_3_ (10 µmol/L) and 0.25 µL H_3_BO_3_ (10 µmol/L) were applied in the solution to create a treatment with sufficient B and a treatment with deficient B with three replicates, respectively. The seedlings were grown in an illuminated culture room at 20 °C under a 16 h light/8 h dark cycle. The plants were arranged on a plant tray that had 9 × 6 holes with 5.6 cm between plants. The nutrient solution was oxygenated by an air pump and was replaced weekly. The pH of the nutrient solution was maintained at 5.7–6.0. When the seedlings showed B deficient symptoms (30 days), the root (R), old leaves (OL) and juvenile leaves (JL) of three plants of QY10 and W10 were sampled for RNA extraction, respectively. All samples were immediately frozen in liquid nitrogen and then stored at −80 °C.

### 2.2. RNA-Seq

Total RNA was extracted using an RNeasy Plant Kit (Bioteke RP1202) according to the manufacturer’s instructions. First, 1.5 µg total RNA from each sample were used for preparing RNA-Seq libraries (cDNA libraries). The manufacturer’s protocol (the Illumina Truseq RNA sample prep Kit, Illumina Inc., San Diego, CA, USA) was used to generate sequencing libraries. The PCR amplification products were checked, excised and purified by agarose gel electrophoresis and MinElute PCR Purification Kit (QIAGEN, Hilden, Germany). The final products were loaded onto flow cell channels with a concentration of 2 pM by using 2 × 100 bp pair-end sequencing strategy. The sequencing analysis were conducted in the National Key Laboratory of Crop Genetic Improvement, Huazhong Agricultural University using Illumina Hiseq^TM^ 2000 platform (Illumina Inc., San Diego, CA, USA). All raw data were deposited in the NCBI Sequence Read Archive (Bioproject accession number, PRJNA393069) [31].

### 2.3. Reads Alignment, Transcript Assembly and Junction Prediction

The average reads generated from these RNA-Seq libraries was more than 170 million (Appendix A). All the above sequences were trimmed based on quality (Q ≥ 30) and 68.2% of clean reads were mapped on the reference genome of Darmor-*bzh* (Table 1) using TopHat software (v2.0.12) [32]. The aligned reads were assembled into transcripts using cufflinks software (v2.2.1) [33]. The cuffcompare package with cufflinks suite program was used to identify novel genes and transcripts that are identical to the *B. napus* reference genome. In addition, TopHat software (v2.0.12) was also used to predict the splice junctions, and to identify the known and novel splice junctions according to the gene information with default parameters. In principal component analysis (PCA) of all genes, the replicate samples showed a high similarity with respect to the first two principal components, little variation within group and a good separation of groups (Appendix A).

### 2.4. Detection of Alternative Splicing Events

All RNA CEL files were normalized using the RPKM (Reads Per kb per Million reads) method [34]. Alternative splicing events were identified by the AS transcriptional landscape visualization tool (AStalavista) [35]. Source codes for the AStalavista software (v3.0) are available online at http://genome.crg.es/astalavista/. TopHat (v2.0.12) was used to calculate the AS events numbers. This study focused on four main types of AS: intron retention, exon skipping (cassette exons), alternative 3′ splice site (alternative acceptor) and alternative 5′ splicing site (alternative donor).

### 2.5. Identification of Differential Alternative Splicing Events and Differentially Expressed Genes

Differential alternative splicing (DAS) events were analyzed using the program Replicate Multivariate Analysis of Transcript Splicing (rMATS, v3.0.9) [36]. A T-test was used to calculate *p* value of splicing events with Δ ψ. A stringent threshold, *p* ≤ 0.05 and |Δ ψ| ≥ 0.05, was used to define the DAS events of AS events between different cultivars or different tissues of oilseed rape. The events with *p* values less than 0.05 were identified as significantly different events. Differentially expressed genes (DEGs) were calculated using a *t*-test (corrected by Benjamin–Hochberg false discovery rate (FDR) multiple testing). Probe-sets with a FDR corrected *p*-value ≤ 0.05 and fold change of >2 were considered to be differentially expressed.

### 2.6. Semi-Quantitative RT-PCR Analysis

Total RNA was extracted using an RNeasy Plant Kit (Bioteke RP1202) according to the manufacturer’s instructions. The DNAase treatment is mentioned above. Synthesizing the cDNA used a reaction solution containing 1 µg RNA with HiFiScript, dNTP Mix, Primer Mix, 5 × RT Buffer, DTT and HiFiScript. The solution was incubated at 42 °C for 50 min, followed by 85 °C for 5 min. The cDNA was used as a PCR template in a 20 µL reaction system of semi-quantitative RT-PCR. To validate the DAS events, 12 pairs of primers were designed based on different DAS events including intron retention, alternative 3′ splice site and alternative 5′ splice site. The primers used are listed in Appendix A.

### 2.7. Gene Ontology (GO) Analysis and KEGG Analysis

Blast2GO was used to determine the gene functional category with a cut-off of 1E-5 [37]. An internal Perl script was used to perform GO annotation based on Open Biological and Biomedical Ontologies (OBO) (http://purl.obolibrary.org/obo). Biochemical pathway was analyzed by Kyoto Encyclopedia of Genes and Genomes (KEGG) Orthology based on annotation system version 2.0 (KOBAS) [38]. The different pathways shown in the figures were chosen based on statistical significance (*p* < 0.05).

## 3. Results

### 3.1. Novel Transcripts and Novel Genes in B. napus

Assembly of QY10 and W10 under B sufficient and deficient conditions identified 135,036 expressed transcripts and 56,203 expressed genes, and each gene had two or more transcripts on average (Table 2). Many new transcripts and genes were discovered by comparing the transcripts of assembling and the reference genome. The numbers of novel isoforms were significantly higher than that of the reference isoforms in both QY10 and W10 whether under B sufficient or deficient conditions (Figure 1). GO analysis of the novel genes revealed that the categories of “ATP hydrolysis coupled proton transport” in biological process, “proton-transporting ATPase activity” in molecular function and “proton-transporting V-type ATPase, V0 domain” in cellular component were significantly enriched for common novel genes in QY10 and W10 under both B conditions (Figure 2). Change of ATP activity could affect the stability of cell membranes by changing the proton gradient on the plasma membrane at low B.

### 3.2. Identification of AS Events in QY10 and W10 under B Sufficient and Deficient Conditions

In total, 11,040,252 splice junctions were identified in all 36 RNA-Seq libraries (Appendix A). Four major types of AS events, namely exon skipping, alternative 5′ splice site, alternative 3′ splice site and intron retention, were investigated in this study (Figure 3). Under B sufficient conditions, the numbers of AS events and genes in the root, juvenile leaves and old leaves of W10 were higher than those of QY10 (Figure 4). The numbers of AS events and genes in root and juvenile leaves of W10 were also higher than those of QY10 under low B conditions (Figure 4). Moreover, the numbers of AS events and genes in both QY10 and W10 under B deficient conditions were higher than those under B sufficient conditions.

The AS genes with the types of splicing of “intron retentions (RI)” and “alternative 3′ splice site (A3SS)” were the largest fraction, whereas the AS genes with “exon skipping (ES)” were the least fraction among all the AS genes in both cultivars whether under B sufficient or B deficient conditions (Figure 5). The number of AS genes with “alternative 5′ splice site (A5SS)” was between the types of “RI”, “A3SS” and “ES”. Most AS genes had one type of AS in the root, juvenile leaves and old leaves in W10 and QY10 under B sufficient and deficient conditions (Figure 6 and Appendix A). Only a small portion of AS genes had all four types of AS in the same organ across two cultivars and two B treatments (Figure 6 and Appendix A). The number of AS genes with the four types of AS in the juvenile leaves of W10 under B deficient condition was the most and that in the juvenile leaves of QY10 under B sufficient condition was the least. The number of AS genes with both “RI” and “A3SS” was the greatest and that of AS genes with both “ES” and “A5SS” was the least in each part investigated (Figure 6 and Appendix A).

The categories of “cell wall modification”, “glucose-6-phosphate transport”, “ion transport”, and “response to stress” were mostly enriched for AS genes, while the categories of “DNA packaging”, “ribosomal export” and “protein location” were enriched for no AS genes (Figure 7). These demonstrated that the genes with AS and non-AS had different functions. Moreover, the AS genes of QY10 and W10 identified in B deficient conditions showed more than that in B sufficient conditions. The AS of most of the genes tended to occur under abiotic stress conditions (Figure 7).

### 3.3. Differential Alternative Splicing in QY10 and W10 in B Deficient Conditions

In total, 159, 190 and 163 DAS genes were identified in root, juvenile leaves and old leaves of QY10, respectively, under B deficient conditions (Figure 8). Among them, 33 DAS genes were detected simultaneously in root, juvenile leaves and old leaves, and were involved in several important biological processes, such as “Phosphoglucomutase”, “Translation elongation factor EF1B”, “glutathione S-transferase phi 8” and “calcineurin B-like protein 9” (Table 3). In total, 24, 40 and 74 DAS genes were distinguished in root, juvenile leaves and old leaves in W10, respectively, in B deficient condition. Only two DAS genes coexisted in root, juvenile leaves and old leaves (Figure 8). They functioned as cystatin and phosphoribulokinase, respectively (Table 3). A total of 63 DAS genes were detected simultaneously in all three organs of QY10 and W10 under B deficient conditions. Moreover, 313 and 58 cultivar-specific DAS genes were identified in QY10 and W10, respectively (Figure 8).

The percentages of the four major types of DAS genes in root, juvenile leaves and old leaves of QY10 under B deficiency, and juvenile leaves and old leaves of W10 under B deficiency, all showed that RI > A3SS, A5SS > ES. However, in the root of W10, the percentage of “ES” DAS genes was the highest, and that of “A5SS” DAS genes was the least among the four major types of DAS genes (Figure 9). If we used W10 as control, the total percentages of “RI” and “A3SS” DAS genes were the highest, whereas that of “ES” DAS genes was the least in QY10 under B sufficient or deficient conditions.

KEGG pathways showed that these DAS genes were involved in all kinds of stress response pathways in *B. napus*, such as “Glycolysis/Gluconeogenesis”, “Calcium signaling pathway”, “MAPK signaling pathway” and “Peroxisome” (Table 4). Some important DAS genes were involved in the B deficient responses, especially in the root in both QY10 and W10. For example, *BnaC05g18490D* was associated with the Glycolysis process, which encoded the phosphoglucomutase and regulated cell wall synthesis. B deficiency led to the increase in retention of intron 13 in the juvenile leaves and old leaves in QY10, and the decrease in retention of intron 13 in the old leaves in W10. In addition, *BnaC07g18010D* and *BnaA02g29830D* were involved in the cell wall synthesis and integrity of cell membrane, which is an important biological process in response to B deficiency. The two genes showed decreased intron retention in old leaves in W10 in B deficient conditions. The results of qRT-PCR of these genes are consistent with the RNA-Seq data (Figure 10). The change of pre-mRNA splicing could influence the metabolic processes in QY10 and W10 in response to B deficiency.

Most of the intron retention transcripts that had a premature termination codon (PTC) are degraded by the “nonsense-mediated mRNA decay (NMD)” surveillance pathway or are targeted by the microRNAs [12,14]. Sequence analysis of the intron retention transcripts of QY10 and W10 at low B showed that PTCs were found in all of these transcripts. The decrease or increase in retention of intron could increase or decrease the abundance of the functional transcripts, respectively. For example, the functional transcript levels of *BnaC05g18490D*, *BnaC07g18010D* and *BnaA02g29830D* were up-regulated in W10 under B deficient condition. In contrast, the functional transcript levels of these genes were down-regulated or unchanged in QY10 under B deplete condition.

### 3.4. DAS Genes and DE Genes in QY10 and W10 in B Deficient Conditions

Combined analysis of DE genes and DAS genes in root, juvenile leaves and old leaves of QY10 and W10 indicated that about 0.06% of DE genes showed AS (Table 5). Under B deficient conditions, the number of genes with both DAS and DE in the root of QY10 was the highest (32), and in the juvenile leaves of W10 was the least (13) (Table 5). If W10 were used as a control, the number of genes with both DAS and DE in the old leaves of QY10 under B sufficient conditions was the highest (31). No gene showed both DAS and DE in the juvenile leaves and old leaves of QY10 under B deplete conditions (Table 5). The number of genes with both *DAS* and *DE* was lower than that of the genes that were either with DAS or DE in QY10 and W10, which demonstrated that the DAS and DE genes were independent.

Functional categorization of the DAS and DE genes in the root, juvenile leaves and old leaves of QY10 and W10 under B deficient conditions revealed that most of DAS and DE genes enriched in different functional pathways in biological process. For example, the categories of “mRNA processing”, “Succinyl–CoA metabolic process”, “signal transduction” and “cellular response to stress” were only enriched for DAS genes, whereas the categories of “cell wall formation”, “cell wall pectin metabolic process” and “cell wall organization” were only enriched for DE genes (Figure 11). Only a small group of DAS and DE genes were annotated to the same function. For example, the categories of “starch metabolism process” and “glucose metabolism process” were enriched for both DAS genes and DE genes (Figure 11).

The majority of the DAS genes had differential intron retention (DIR), such as *BnaA07g33860D* (Sulphate anion transporter); and alternative 3′ splice site (DA3SS), such as *BnaA01g30340D* (SANT/Myb domain) and *BnaA06g17710D* (alpha-glucan phosphorylase 2, PHS2) (Appendix A). These genes also showed significant up-regulation or down-regulation under B deplete conditions (Appendix A).

### 3.5. SR Splicing Factors in QY10 and W10 under B Deficient Condition

SR splicing factors play a crucial role to regulate pre-mRNA splicing in plants. A total of 34 SR splicing factors were identified in QY10 and W10 under B deficient conditions (Appendix A). Four of these SR splicing factors, *BnaC06g14780D*, *BnaA01g14750D*, *BnaA06g15930D* and *BnaC01g41640D*, underwent DAS. Under B deplete conditions, A3SS occurred in the genes of *BnaC06g14780D* in the root and juvenile leaves; A5SS occurred in the genes of *BnaA01g14750D* in old leaves of QY10; and ES occurred in the gene of *BnaC01g41640D* in the root and old leaves of QY10 and *BnaA06g15930D* in the juvenile leaves of W10 (Appendix A).

*BnaC06g14780D* showed differential A3SS in the second intron of the root and juvenile leaves in QY10 under different B conditions (Figure 12). In addition, its expression level under B deficient conditions was lower than that in B sufficient conditions. Compared with the *B. napus* reference genome of Darmor-bzh, the A3SS imported a novel sequence. The exon insertion of *BnaC06g14780D* in the root and juvenile leaves in QY10 could increase the abundance of functional transcript in response to B deficiency (Figure 12). *BnaA01g14750D* had lower A5SS in intron 2 in the root of QY10 under B deficient conditions than B sufficient conditions. A5SS also produced a novel sequence with PTC (Figure 12). The exon insertion of *BnaA01g14750D* in the root of QY10 could also increase the abundance of the functional transcript under B deficient conditions. In addition, *BnaA06g15930D* and *BnaC01g41640D* were exon skipping, which constituted only a small portion of differentially expressed alternatively spliced genes in plants (data not shown).

*BnaC06g14780D* was a seed gene, and its downstream target genes were associated with the function of B [18], such as expansion protein (*BnaA09g52970D*) and bZIP transcription factor (*BnaC04g52770D*) (Table 6 and Appendix A).

## 4. Discussion

Alternative splicing, which generates multiple transcripts from the same gene, is an important modulator of gene expression that can increase proteome diversity and regulate mRNA levels [15]. RNA-Seq data revealed AS in 48% of genes in *B. napus* cultivar “Darmor-bzh” [39]. In this study, 30% and 35% of intron-containing genes underwent AS in QY10 and W10 under B sufficient and deficient conditions, respectively (Appendix A). The number of AS gene identified in this study was lower than that in the cultivar “Darmor-bzh”, possibly because the sequencing depth of the former is lower than the latter. The increase in AS genes of *B. napus* under B deficient conditions (Figure 4) indicated that AS might be an important strategy of posttranscriptional regulation, and the increase in AS could improve the molecular plasticity of plants to adapt to abiotic stress. Recently, a higher proportion of genes are detected showing AS under salt stress in Arabidopsis [29], drought stress in maize [40], heat shock in the moss *Physcomitrella patens* [28] and high temperature in grape [11]. AS regulates the plant response to abiotic stress are largely by targeting the abscisisc acid (ABA) pathway [15]. However, the AS genes of QY10 and W10 identified in B deficient conditions were not associated with the ABA signaling (Figure 7).

In this study, the number of AS genes with the types of “RI”, “A3SS” and “A5SS” were much more than that with “ES” and others in *B. napus*, and RI was the most prominent types of AS (Figure 5). In *B. napus* cultivar “Darmor-bzh”, intron retention is also frequent (62%), whereas exon skipping is rare (3%) [39]. High proportions of AS genes with the types of “RI” are also found in maize [41] and *Physcomitrella patens* [42]. Although the intron retention is highly repressed by elevated temperature in *Physcomitrella patens*, the AS genes with the types of “RI” constituted the largest fraction of alternatively spliced genes [28]. Moreover, the genes responded to B deficient condition in this study, such as *BnaC05g18490D*, *BnaC07g18010D* and *BnaA02g29830D,* which were associated with cell wall synthesis and integrity of cell membrane, showed differential RI during B deficiency (Appendix A). AS events based on different splicing types may lead to functionally relevant changes in the protein products [43]. For example, 4% of R2R3-MYB genes had undergone AS events in soybean, which generate a variety of transcripts to increase the complexity of transcriptome [44]. In this study, the categories of “cell wall modification”, “glucose-6-phosphate transport”, “ion transport”, and “response to stress” were mostly enriched for AS genes, while the categories of “DNA packaging”, “ribosomal export” and “protein location” were enriched for non-AS genes (Figure 7), which indicated that the genes with AS and non-AS play distinct physiological roles.

*B. napus* is highly susceptible to B deficiency [20]. There are significant genotypic differences in the response to low-B stress among different *B. napus* cultivars [21]. Under B deplete conditions, 159, 190 and 163 DAS genes were identified in the root, juvenile leaves and old leaves of QY10 (B-efficient cultivar), respectively; however, only 24, 40 and 74 DAS genes were identified in the root, juvenile leaves and old leaves of W10 (B-inefficient cultivar) (Figure 8). The increase of AS events occurred under abiotic stress conditions could enhance the tolerance ability of plants [15]. Further experiments should be conducted to confirm whether the tolerance ability of QY10 was higher than that of W10 belong to the DAS genes generated in B-efficient cultivar (QY10) were far more than that in B-inefficient cultivar (W10) under B deplete conditions.

The number of genes with both DAS and DE were far lower than that of the genes that were either DAS or DE in QY10 and W10 (Table 5), which demonstrated that the DAS and DE genes were independent. Functional categorization of DAS and DE genes of root, juvenile leaves and old leaves in QY10 and W10 under B deficient conditions revealed that most of DAS and DE genes enriched in differential functional pathways in biological process. However, only a small group of DAS and DE genes were annotated to the same function (Table 5, Figure 11). The majority of the DAS genes had differential intron retention (DIR) and alternative 3′ splice site (DA3SS), which also showed significant up-regulation or down-regulation in B deplete condition (Appendix A). These results demonstrate that both transcriptional regulation and posttranscriptional regulation contribute to *B. napus* adaption to B deficiency.

SR splicing factors play a crucial role in regulation of pre-mRNA splicing in plants [45]. In Arabidopsis, the AS pattern of members of SR splicing factors have been shown to change under various stress conditions, such as high light intensity, salinity and temperature stress [46,47,48]. This suggests that the stress-induced changes in SR splicing factors could in turn alter the factors of downstream targets in response to stress environments [48]. In this study, 34 SR splicing factors in QY10 and W10 were found under B deficient conditions (Appendix A). Four of them, namely *BnaC06g14780D*, *BnaA01g14750D*, *BnaA06g15930D* and *BnaC01g41640D,* showed differential alternative spliced genes (Appendix A). The decrease of novel sequence of *BnaC06g14780D* and *BnaA01g14750D* in the root of QY10 under B deplete conditions suggested that the increase of functional isoforms in the two SR genes might improve plant tolerance in response to B deficiency.

## 5. Conclusions

A total of ~33.3% genes showed AS in *B. napus* cultivars QY10 and W10. The AS genes with the types of splicing of “intron retentions (RI)” and “alternative 3′ splice site (A3SS)” in both cultivars showed the largest fraction whether under B deficient or sufficient conditions. Further experiments should be conducted to confirm whether the tolerance ability of QY10 was higher than that of W10 was attributed to far more DAS genes were identified in QY10 under low B conditions than in W10. To find the functional transcript of the genes, such as SR splicing factor BnaC06g14780D, which responds to low boron stress, would provide a new way to increase the ability of *B. napus* to cope with B deplete stress.

## Figures and Tables

**Figure 1 genes-10-00224-f001:**
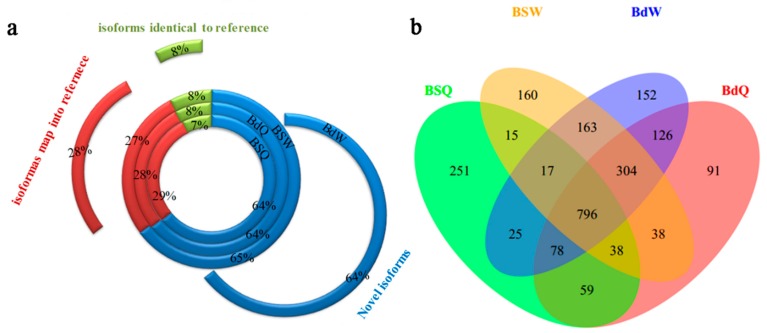
The proportion of assembled transcripts and number of novel genes in the root, old leaves and juvenile leaves of *B. napus* cultivars Qingyou10 (QY10) and Westar10 (W10) under boron (B) sufficient and deficient conditions: (**a**) the proportion of novel isoforms, the isoforms mapped to the reference, and the isoforms identical to the reference; and (**b**) Venn diagram of the overlap of the novel genes. Q, QY, Qingyou10; W, W10, Westar10; Bs, B sufficient condition; Bd, B deficient condition.

**Figure 2 genes-10-00224-f002:**
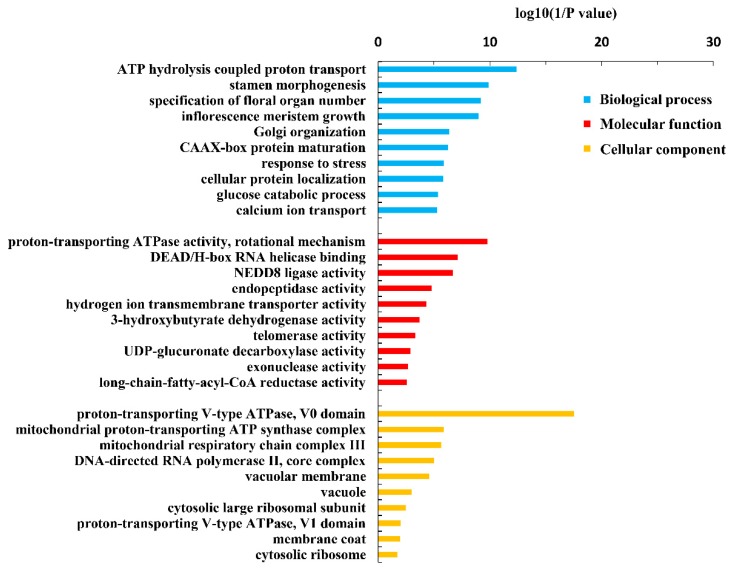
GO terms of the functional categorization of the same novel genes in the root, old leaves and juvenile leaves of *B. napus* cultivars Qingyou10 (QY10) or Westar10 (W10) under boron sufficient and deficient conditions.

**Figure 3 genes-10-00224-f003:**
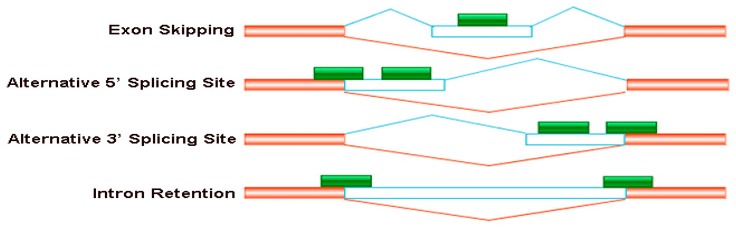
Four major types of the alternative splicing events occurred in *B. napus* cultivars Qingyou10 and Westar10. Exon skipping (ES), alternative 5′ splice site (A5SS), alternative 3′ splice site (A3SS), and intron retention (RI) are shown schematically. The orange/transparent boxes represent exons, green boxes represent sequencing reads and lines represent introns.

**Figure 4 genes-10-00224-f004:**
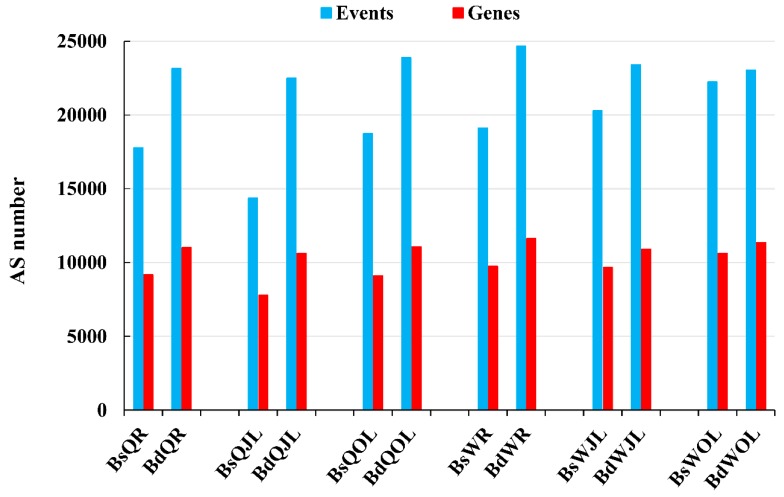
The numbers of the alternative splicing (AS) events and genes in *B. napus* cultivars Qingyou10 (QY10) and Wesatr10 (W10) under boron (B) sufficient and deficient conditions. Q, QY10, Qingyou10; W, W10, Westar10; Bs, B sufficient condition; Bd, B deficient condition; R, root; JL, juvenile leaves; OL, old leaves.

**Figure 5 genes-10-00224-f005:**
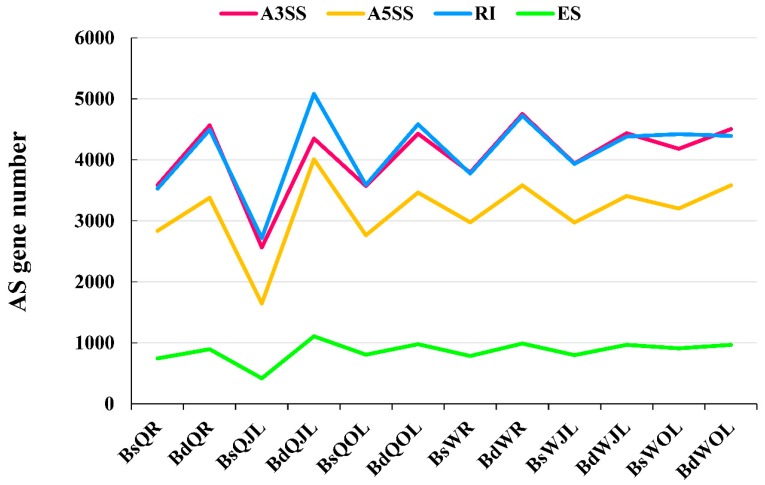
The number of genes of four types of alternative splicing (AS) in *B. napus* cultivars Qingyou10 (QY10) and Wesatr10 (W10) under boron (B) sufficient and deficient conditions. Q, QY10, QY10, Qingyou10; W, W10, Westar10; Bs, B sufficient condition; Bd, B deficient condition; R, root; JL, juvenile leaves; OL, old leaves; A3SS, alternative 3′ splice site; A5SS, alternative 5′ splice site; RI, retain intron; ES, exon skip.

**Figure 6 genes-10-00224-f006:**
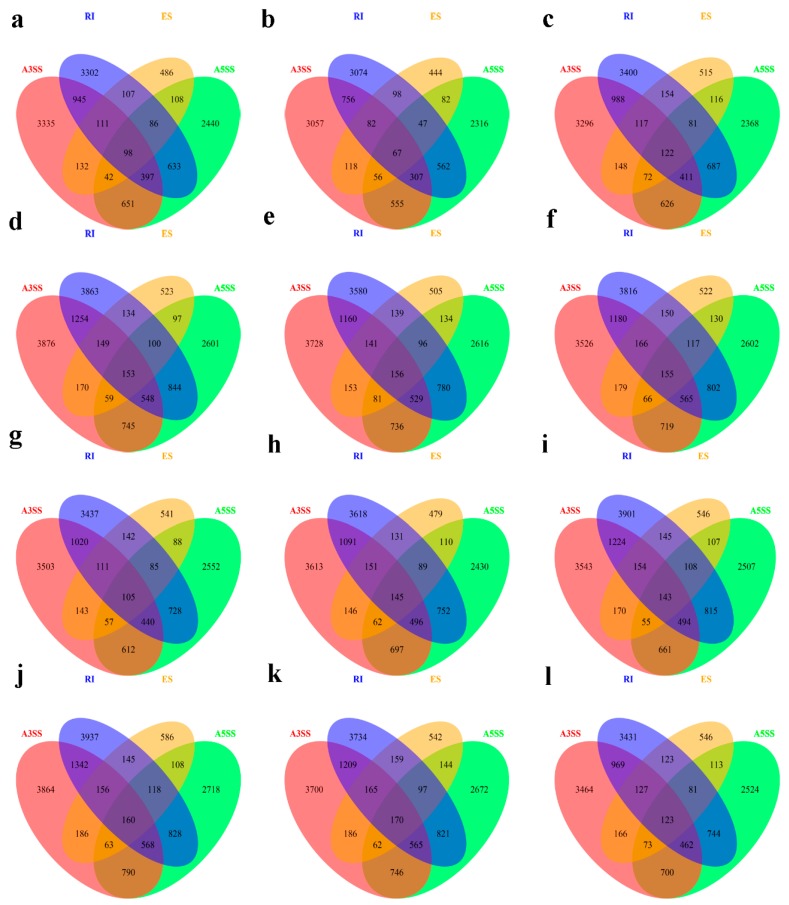
Venn diagram of the overlap of the four types of AS genes in *B. napus* cultivars Qingyou10 (QY10) and Westar10 (W10) under boron (B) sufficient and deficient conditions: (**a**) BsQR; (**b**) BsQJL; (**c**) BsQOL; (**d**) BdQR; (**e**) BdQJL; (**f**) BdQOL; (**g**) BsWR; (**h**) BsWJL; (**i**) BsWOL; (**j**) BdWR; (**k**) BdWJL; and (**l**) BdWOL. Q, QY10, Qingyou10; W, W10, Westar10; R, root; JL, juvenile leaves; OL, old leaves; Bs, B sufficient condition; Bd, B deficient condition.

**Figure 7 genes-10-00224-f007:**
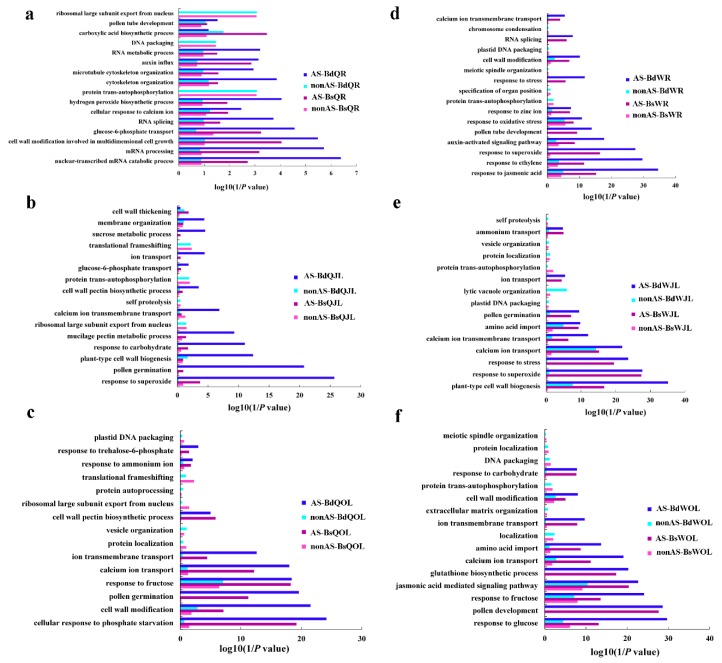
GO terms of the functional categorization of AS and non-alternative splicing (nonAS) genes (biological process) in the root, juvenile leaves and old leaves of *B. napus* cultivars Qingyou10 (QY10) and Westar10 (W10) under boron (B) sufficient and deficient conditions: (**a**) root of QY10; (**b**) juvenile leaves of QY10; (**c**) old leaves of QY10; (**d**) root of W10; (**e**) juvenile leaves of W10; and (**f**) old leaves of W10. AS, alternative splicing; nonAS, non-alternative splicing; Bs, B sufficient condition; Bd, B deficient condition; QR, root of QY10; QOL, old leaves of QY10; QJL, juvenile leaves of QY10; WR, root of W10; WOL, old leaves of W10; WJL, juvenile leaves of W10.

**Figure 8 genes-10-00224-f008:**
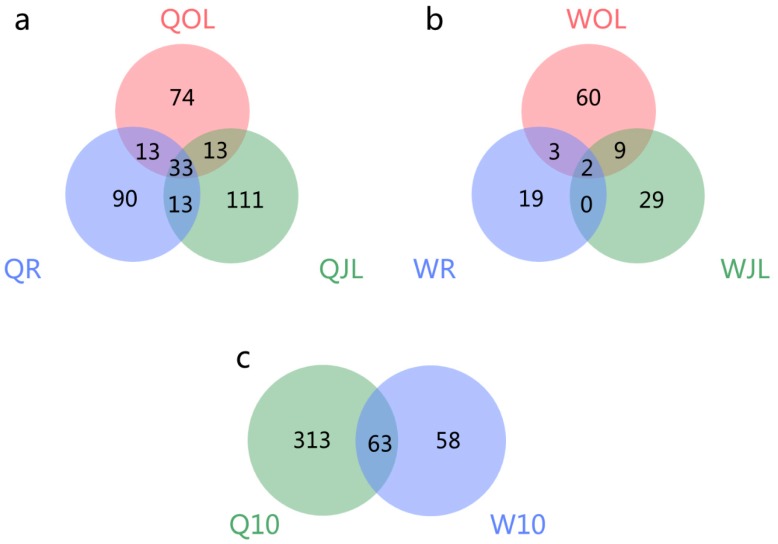
Venn diagram of the overlap of differential alternative splicing (DAS) genes in the root, juvenile leaves and old leaves of *B. napus* cultivars Qingyou10 (QY10) and Westar10 (W10) under boron deficient conditions: (**a**) QY10; (**b**)W10; and (**c**) QY10 and W10 under B deficient conditions; QR, root of QY10; QOL, old leaves of QY10; QJL, juvenile leaves of QY10; WR, root of W10; WOL, old leaves of W10; WJL, juvenile leaves of W10.

**Figure 9 genes-10-00224-f009:**
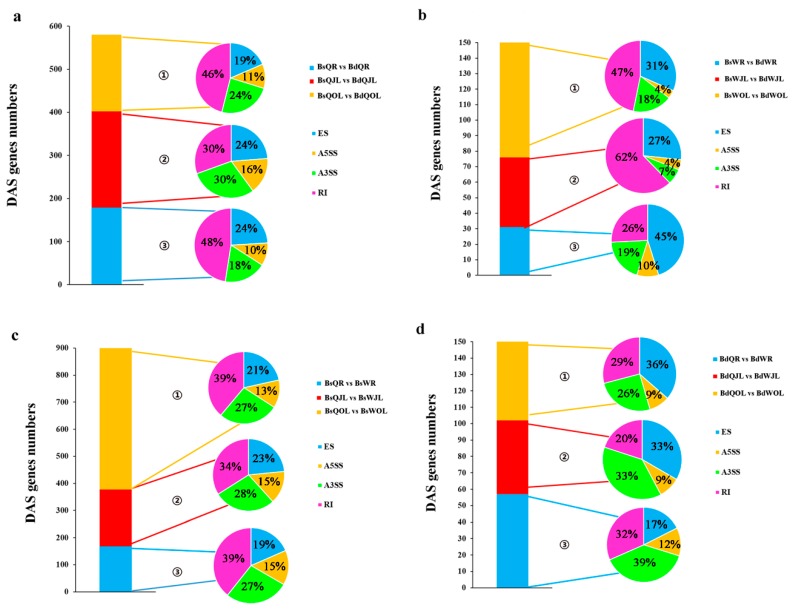
The number of DAS genes in the root, juvenile leaves and old leaves of *B. napus* cultivars Qingyou10 (QY10) and Westar10 (W10) under boron deficient conditions: (**a**,**b**) the total number of DAS genes (control, B sufficient condition) identified in QY10 and W10, respectively; and (**c**,**d**), the total number of DAS genes in QY10 (control, W10) under boron sufficient and deficient conditions, respectively. ① Ratio of four alternative splicing patterns in old leaves; ② ratio of four alternative splicing patterns in juvenile leaves; and ③ ratio of four alternative splicing patterns in root. DAS, differential alternative splicing; Bs, boron sufficient condition; Bd, boron deficient condition; QR, root of QY10; QOL, old leaves of QY10; QJL, juvenile leaves of QY10; WR, root of W10; WOL, old leaves of W10; WJL, juvenile leaves of W10. A3SS, Alternative 3′ splice site; A5SS, alternative 5′ splice site; RI, retain intron; ES, exon skip.

**Figure 10 genes-10-00224-f010:**
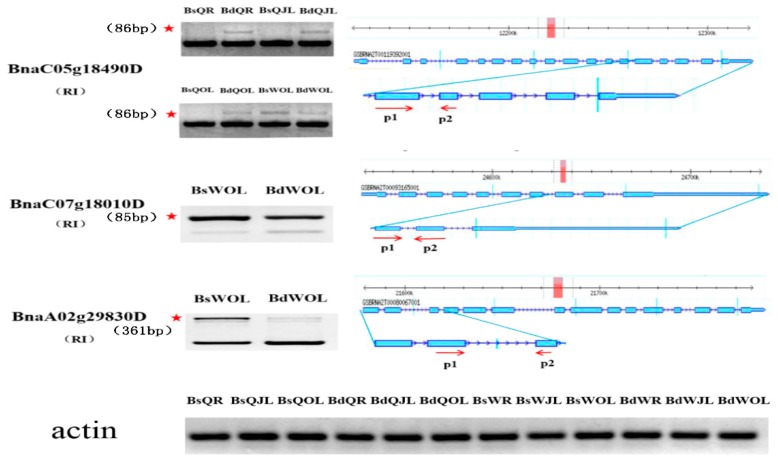
Validation of DAS events in the root, juvenile leaves and old leaves of *B. napus* cultivars Qingyou10 (QY10) and Westar10 (W10) under boron deficient conditions. The band with the red asterisk showed the fragment generated by the alternative splicing of the gene, which was determined according to the size of the retained intron fragment. The forward and reverse primers, P1 and P2, were designed based on the exon between the retained intron. Bs, boron sufficient condition; Bd, boron deficient condition; QR, root of QY10; QOL, old leaves of QY10; QJL, juvenile leaves of QY10; WR, root of W10; WOL, old leaves of W10; WJL, juvenile leaves of W10. RI, retain intron.

**Figure 11 genes-10-00224-f011:**
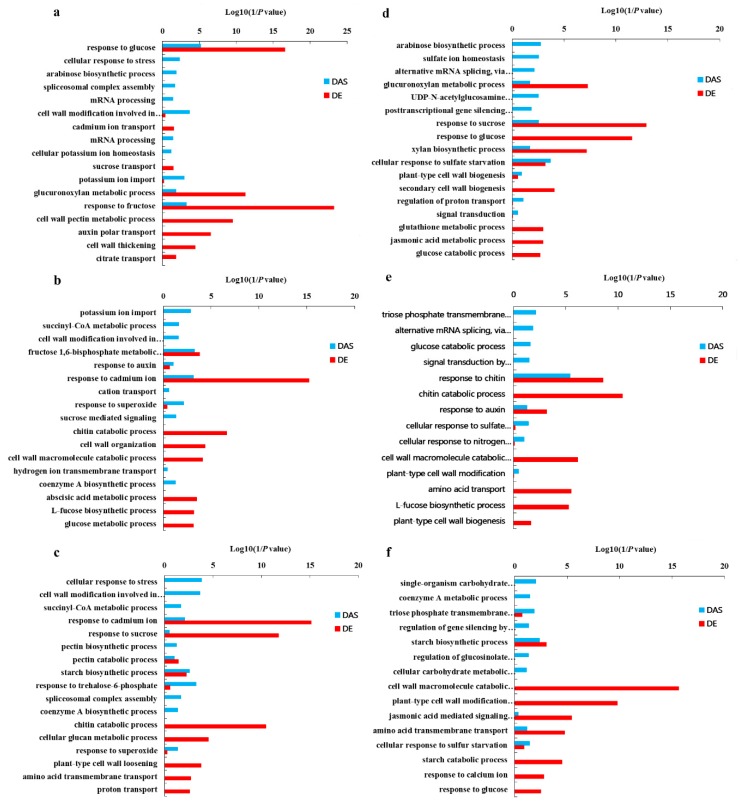
GO terms of the functional categorization of DAS and DE genes (biological process) in the root, juvenile leaves and old leaves of *B. napus* cultivars Qingyou10 (QY10) and Westar10 (W10) under B deficient conditions: (**a**) root of QY10; (**b**) juvenile leaves of QY10; (**c**) old leaves of QY10; (**d**) root of W10; (**e**) juvenile leaves of W10; and (**f**) old leaves of W10.

**Figure 12 genes-10-00224-f012:**
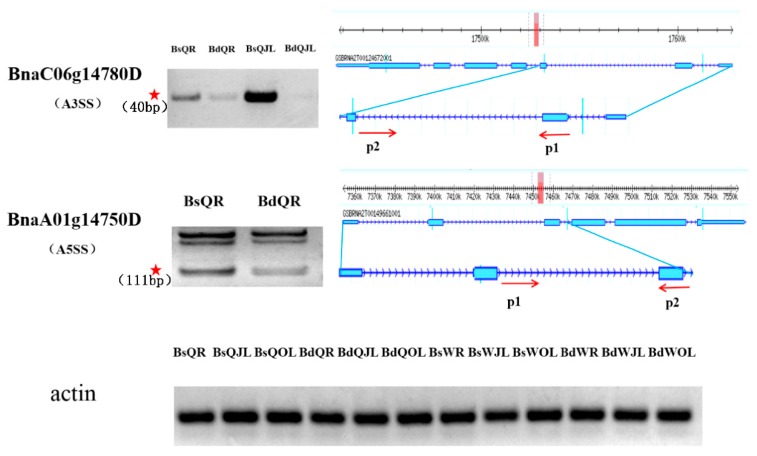
Validation of differential alternative splicing factors in the root and juvenile leaves of *B. napus* cultivars Qingyou10 (QY10) and Westar10 (W10) under boron deficient conditions. The band with the red asterisk show the fragment generated by the alternative splicing of the gene, which was determined according to the size of the insert fragment. The forward and reverse primers, P1 and P2, were used to amplify the insert fragment and side exon. Bs, boron sufficient condition; Bd, boron deficient condition; QR, root of QY10; QOL, old leaves of QY10; QJL, juvenile leaves of QY10; WR, root of W10; WOL, old leaves of W10; WJL, juvenile leaves of W10. A3SS, alternative 3′ splicing site; A5SS, alternative 5′ splicing site.

**Table 1 genes-10-00224-t001:** The starting reads, clean reads ratio, and mapped reads ratio against in the *B. napus* reference genome of *c.v. Darmor-bzh*.

	No. of Reads (Millions)	Reads Percentage (%)
Starting reads	2078.8	-
Clean reads	1921.9	92.45
Mapped reads	1310.9	68.2

**Table 2 genes-10-00224-t002:** The number of novel transcripts and novel genes in the root, juvenile leaves and old leaves of Qingyou10 and Westar10 under B sufficient and deficient conditions (above 1.0 FPKM).

Sample Name	Novel Transcripts	Reference Like Transcripts	Novel Genes	Reference Like Genes
BsQR1	32,762	16,656	1019	5877
BsQR2	33,451	16,384	1136	5766
BsQR3	33,662	16,661	1096	5841
BsQJL1	33,763	15,740	1147	5539
BsQJL2	35,091	15,637	1248	5742
BsQJL3	33,274	15,608	1221	5400
BsQOL1	34,228	14,921	1288	5033
BsQOL2	34,501	14,959	1299	5074
BsQOL3	33,610	14,592	1330	4927
BdQR1	35,565	16,441	1415	5648
BdQR2	35,780	16,570	1549	5643
BdQR3	36,428	16,731	1545	5658
BdQJL1	37,831	16,365	1601	5455
BdQJL2	36,892	16,614	1518	5625
BdQJL3	37,677	16,124	1500	5391
BdQOL1	37,624	16,125	1685	5291
BdQOL2	35,766	16,028	1473	5339
BdQOL3	36,421	16,189	1513	5383
BsWR1	34,561	16,128	1357	5563
BsWR2	35,245	16,801	1433	5777
BsWR3	34,940	17,124	1417	5895
BsWJL1	36,579	16,029	1513	5412
BsWJL2	38,206	16,252	1470	5382
BsWJL3	37,242	16,214	1529	5468
BsWOL1	38,490	15,781	1812	5114
BsWOL2	37,855	15,850	1767	5161
BsWOL3	37,224	15,411	1713	5025
BdWR1	35,174	16,653	1407	5738
BdWR2	35,790	17,013	1483	5830
BdWR3	36,108	16,838	1494	5766
BdWJL1	36,001	15,875	1498	5328
BdWJL2	37,751	16,557	1588	5538
BdWJL3	36,854	15,976	1523	5360
BdWOL1	36,841	16,041	1559	5325
BdWOL2	36,942	15,973	1485	5335
BdWOL3	35,714	15,746	1517	5284

Note: Bs, B sufficient condition; Bd, B deficient condition; Q, QY10, Qingyou10, B-efficient cultivar; W, W10, Westar10, B-inefficient cultivar; R, root; JL, juvenile leaf; OL, old leaf; 1, 2 and 3 indicate Replication 1, Replication 2, and Replication 3, respectively.

**Table 3 genes-10-00224-t003:** Thirty-three and two differential alternative splicing genes detected in root, juvenile leaves and old leaves simultaneously in *B. napus* cultivars Qingyou10 and Westar10, respectively.

Cultivars	GeneID	Gene Description	DAS-Type
QY10	*BnaC09g39180D*	alpha/beta-Hydrolases superfamily protein	A5SS
	*BnaA06g17710D*	alpha-glucan phosphorylase 2 (PHS2)	RI
	*BnaC01g05800D*	AME3	RI
	*BnaC03g31650D*	ATOZI1	A3SS
	*BnaC09g20000D*	calcineurin B-like protein 9 (CBL9)	ES
	*BnaC06g30700D*	cAMP-regulated phosphoprotein 19-related protein	A3SS
	*BnaC05g02560D*	casein kinase like 13 (CKL13)	A5SS
	*BnaC08g08360D*	CONSTITUTIVE PHOTOMORPHOGENIC 9 (COP9)	RI
	*BnaA03g27430D*	*DE*AD box RNA helicase family protein	RI
	*BnaA09g27350D*	dormancy-associated protein-like 1 (DYL1)	RI
	*BnaA03g39560D*	enhancer of ag-4 2 (hua2)	RI
	*BnaC03g57920D*	FK506 binding protein 53 (fkbp53)	A3SS
	*BnaAnng37730D*	glutathione S-transferase phi 8 (GSTF8)	ES
	*BnaA07g00230D*	actin 7 (ACT7)	RI
	*BnaA04g12350D*	glycine rich protein 7 (atgrp7)	A5SS
	*BnaC04g04470D*	homeodomain GLABROUS 1 (HDG1)	RI
	*BnaA06g30540D*	NADH-ubiquinone oxidoreductase B8 subunit, putative	ES
	*BnaA06g38980D*	nitrilase 2 (NIT2)	ES
	*BnaC09g20910D*	peptide transporter 2 (PTR2)	RI
	*BnaC05g18490D*	Phosphoglucomutase/phosphomannomutase family protein	RI
	*BnaA07g16600D*	PUR5	A3SS
	*BnaC05g24350D*	radical-induced cell death1 (rcd1)	RI
	*BnaA03g30650D*	Ribosomal L29 family protein	ES
	*BnaC09g54460D*	Ribosomal protein S13/S18 family	A3SS
	*BnaA07g25750D*	RNA-binding (RRM/RBD/RNP motifs) family protein	RI
	*BnaC06g14780D*	RSZ32	A3SS
	*BnaA07g16660D*	sedoheptulose-bisphosphatase (SBPASE)	ES
	*BnaC05g08610D*	sugar transporter 1 (STP1)	A3SS
	*BnaC04g31660D*	TLD-domain containing nucleolar protein	RI
	*BnaA03g07610D*	Translation elongation factor EF1B	A3SS
	*BnaC04g56630D*	unknown protein	RI
	*BnaCnng63660D*	unknown protein	RI
	*BnaC03g03780D*	VND-interacting 1 (VNI1)	A3SS
W10	*BnaCnng40950D*	Cystatin/monellin superfamily protein	A3SS
	*BnaC07g51220D*	nicotinate phosphoribosyltransferase 1 (NAPRT1)	A5SS

Note: DAS, Differential alternative splicing; R, root; JL, juvenile leaves; OL, old leaves; A3SS, Alternative 3′ splice site; A5SS, Alternative 5′ splice site; RI, retain intron; ES, exon skip.

**Table 4 genes-10-00224-t004:** KEGG pathways of differential alternative splicing genes in root, juvenile leaves and old leaves of *B. napus* cultivars Qingyou10 and Westar10 under B deficient conditions.

Tissues	KEGG Pathways	Pathway ID	No. of DAS Gene	*p*-Value
QR	Spliceosome	ko03040	13	3.08E-06
	Carbon metabolism	ko01200	7	0.034913379
	Carbon fixation in photosynthetic organisms	ko00710	3	0.034913379
	Biosynthesis of amino acids	ko01230	6	0.034913379
	mRNA surveillance pathway	ko03015	4	0.034913379
	Glycolysis / Gluconeogenesis	ko00010	3	0.034913379
	Citrate cycle (TCA cycle)	ko00020	2	0.034913379
	Glyoxylate and dicarboxylate metabolism	ko00630	2	0.034913379
	Pyruvate metabolism	ko00620	2	0.034913379
	Sulfur metabolism	ko00920	1	0.034913379
	Nitrogen metabolism	ko00910	1	0.034913379
	Calcium signaling pathway	ko04020	1	0.034913379
	Galactose metabolism	ko00052	1	0.035225944
	MAPK signaling pathway	ko04010	1	0.035225944
	RNA transport	ko03013	2	0.037548325
	Arginine and proline metabolism	ko00330	1	0.038190214
	Starch and sucrose metabolism	ko00500	2	0.039194028
	Amino sugar and nucleotide sugar metabolism	ko00520	1	0.043604665
	Oxidative phosphorylation	ko00190	1	0.046236768
	Plant hormone signal transduction	ko04075	2	0.047198466
QJL	Carbon fixation in photosynthetic organisms	ko00710	4	0.011345749
	Tryptophan metabolism	ko00380	3	0.011345749
	Pentose phosphate pathway	ko00030	3	0.011345749
	Glycolysis / Gluconeogenesis	ko00010	3	0.019619536
	Spliceosome	ko03040	4	0.019619536
	Glyoxylate and dicarboxylate metabolism	ko00630	2	0.019619536
	Pyruvate metabolism	ko00620	2	0.019619536
	ABC transporters	ko02010	1	0.019619536
	Carbon metabolism	ko01200	4	0.019619536
	Fructose and mannose metabolism	ko00051	1	0.019619536
	MAPK signaling pathway	ko04010	1	0.019619536
	Glycine, serine and threonine metabolism	ko00260	1	0.020680045
	Arginine and proline metabolism	ko00330	1	0.021716355
	Starch and sucrose metabolism	ko00500	2	0.021716355
	Peroxisome	ko04146	1	0.021716355
	Pentose and glucuronate interconversions	ko00040	1	0.021716355
	Glutathione metabolism	ko00480	1	0.021865611
	Amino sugar and nucleotide sugar metabolism	ko00520	1	0.024431115
	Plant hormone signal transduction	ko04075	2	0.026222312
	RNA transport	ko03013	1	0.02673548
	Biosynthesis of amino acids	ko01230	1	0.029409127
QOL	Spliceosome	ko03040	7	0.030261395
	AMPK signaling pathway	ko04152	5	0.030261395
	Pyruvate metabolism	ko00620	3	0.047340426
	Tryptophan metabolism	ko00380	2	0.049084834
WR	Ribosome	ko03010	2	0.000561484
WJL	Peroxisome	ko04146	2	0.027237847
	AMPK signaling pathway	ko04152	2	0.042503393
	Glyoxylate and dicarboxylate metabolism	ko00630	1	0.190757264
	Glycolysis / Gluconeogenesis	ko00010	1	0.278858201
WOL	Starch and sucrose metabolism	ko00500	3	0.034551184
	Amino sugar and nucleotide sugar metabolism	ko00520	2	0.076377304
	Pentose phosphate pathway	ko00030	1	0.165990643
	Galactose metabolism	ko00052	1	0.173014536

Note: DAS, Differential alternative splicing; Bs, B sufficient condition; Bd, B deficient condition; QR, root of QY10; QOL, old leaves of QY10; QJL, juvenile leaves of QY10; WR, root of W10; WOL, old leaves of W10; WJL, juvenile leaves of W10.

**Table 5 genes-10-00224-t005:** The number of differential alternative splicing genes and differential expressed genes in the root, juvenile leaf and old leaf of *B. napus* cultivars Qingyou10 and Westar10 under boron deficient conditions.

Sample Names	DAS Genes	DE Genes	Overlap
BsQR vs. BdQR	179	3404	32
BsQJL vs. BdQJL	223	1482	30
BsQOL vs. BdQOL	178	1364	8
BsWR vs. BdWR	32	2053	7
BsWJL vs. BdWJL	47	1054	1
BsWOL vs. BdWOL	85	1181	13
BsQR vs. BsWR	174	3253	10
BsQJL vs. BsWJL	217	744	4
BsQOL vs. BsWOL	589	2769	31
BdQR vs. BdWR	37	102	3
BdQJL vs. BdWJL	49	51	0
BdQOL vs. BdWOL	62	196	0

Note: DAS, differential alternative splicing; DE, differential expressed; Bs, B sufficient condition; Bd, B deficient condition; QR, root of QY10; QOL, old leaves of QY10; QJL, juvenile leaves of QY10; WR, root of W10; WOL, old leaves of W10; WJL, juvenile leaves of W10.

**Table 6 genes-10-00224-t006:** Different types of target genes and gene interaction in the splicing factor gene network of *B. napus* cultivars Qingyou10 and Westar10 under boron deficient conditions.

Gene Name	Target Genes	Z-Score	Gene Description	DAS Type-Regulation					
				Boron Deficient/Boron Sufficient					
				QR	QJL	QOL	WR	WJL	WOL
BnaC06g14780D	*BnaC04g52770D*	0.92	Basic-leucine zipper (bZIP) transcription factor	RI	-	-	-	-	-
	*BnaC01g37580D*	0.91	Protein kinase domain	ES	-	ES	ES	-	-
	*BnaA09g52970D*	0.88	Expansin	RI	-	-	-	-	-
	*BnaA01g14590D*	0.87	Zinc finger, RING-type	-	-	RI	-	-	-
	*BnaC04g12670D*	0.86	Folate-biopterin transporter	-	A3SS	-	-	-	-
	*BnaA05g30860D*	0.83	Glycosyl hydrolase family 100	-	-	ES	-	-	-
	*BnaC02g31500D*	0.80	Pectinacetylesterase	-	-	-	-	ES	-
	*BnaA07g32180D*	0.80	VPS35 homolog B	RI	-	RI	-	-	-
	*BnaA01g30320D*	0.80	Phosphoglycerate kinase	A3SS	-	-	-	-	-

Note: Z-score indicated the level of target genes interacted with gene; the higher of Z-score, the higher level of target genes interacted with gene; Bs, B sufficient condition; Bd, B deficient condition; QR, root of QY10; QOL, old leaves of QY10; QJL, juvenile leaves of QY10; WR, root of W10; WOL, old leaves of W10; WJL, juvenile leaves of W10; A3SS, alternative 3′ splicing site; A3SS, alternative 3′ splicing site; RI, retain intron; ES, exon skipping; ES, exon skipping; “-”, non-*DAS*.

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
