# Peer review of "Differential Alternative Splicing Genes in Response to Boron Deficiency in Brassica napus"

_genes, 2019, doi:10.3390/genes10030224_

Round 1

Reviewer 1 Report

I encourage the author to continue in this field of research

Reviewer 2 Report

Please find my comments in the attached file.

This manuscript is a resubmission of an earlier submission. The following is a list of the peer review reports and author responses from that submission.

Round 1

Reviewer 1 Report

The authors of the manuscript ID: genes-412108 entitled: ‘Differential alternative splicing genes in response to boron deficiency in Brassica napus’ used a RNA-seq library set which has been published earlier on to elucidate whether changes in alternative splicing do occur between B-efficient (Q10) and B-inefficient (W10) Brassica napus cultivars grown under B-deficient or B-sufficient conditions.

With respect to the presented manuscript, the following points should be considered:

Major Point 1

In Figure 1, Figure 4, Figure 5, Figure 6, Figure 7, Figure 8, Figure 11, Table 3, and Table 5 etc. statistics are missing. The RNA-seq libraries represent three repetitions of the same biological material (2 cultivars x 2 B conditions x 3 tissues x 3 repetitions). The authors must provide statistics for their analyses based on the three repetitions and available RNA-seq libraries. Statistics allow the reader to see whether the observed results are indeed significant differences or whether the results are not significant. Maybe the authors showed only significant results but that must be indicated and stated in the Figures (error bars) and the Figure Legends. Moreover, it should be stated by which statistical test the corresponding analyses were evaluated. It is suggested to upload and link the metafiles of the analyses to the present manuscript

Major Point 2

The conclusions of the manuscript remain unfortunately mostly speculations based on correlations identified. However, experimental effort is missing to actually verify the formulated hypotheses.

Please find the following examples. The authors formulate the following hypothesis:

Example1: ‘The tolerance ability of QY10 was higher than that of W10 might belong to the DAS genes generated at B –efficient cultivar (QY10) were far more than that in B –inefficient cultivar (W10) at B deplete condition.’ There is no experimental result presented, which might support this hypothesis. The fact that QY10 is B efficient and W10 is B inefficient and Q10 has more As than W10 is only correlative and could be formulated for all the other differences which are present between the two cultivars. Either the authors provide an experimental approach to confirm this statement or it should be rephrased.

Example2: ‘The genes responded to B deficient condition, such as BnaC05g18490D, BnaC07g18010D and BnaA02g29830D, which were associated with cell wall synthesis and integrity of cell membrane, showed differential RI during B deficiency (Table S5), which supported that RI was a fine-tune means to regulate the expression of functional transcripts to adapt to B deficiency’.

As reported by the present manuscript, a lot of genes are regulated under B deficient conditions and a lot of genes are not regulated. Some of these genes underwent AS, some did not. Therefore, it is very speculative to conclude, based on very few genes, that RI and AS is a fine-tuning tool to regulate the expression of functional transcripts to adapt to B deficiency’.

If such a conclusion is drawn, experimental evidence should be provided. The ‘weak’ correlation is, to my understanding, not enough to draw such a conclusion. Are there hints which can support that hypothesis already available, for instance in other available datasets on plant material which was grown under plus and minus boron conditions: E.g. GSE52208 available at the Gene Expression Omnibus at NCBI.

Example3: The authors speculate based on their in silico analysis that:

‘These suggested that over-expression of functional isoforms of SR splicing factors in specific tissues could increase the ability of B. napus in response to B deficient condition.’ The authors should test this hypothesis experimentally.

Major Point 3

Figure 12 and Figure10: Size markers are missing in the provided gel pictures, to see whether the generated band size is correlating with the expected band size. Moreover, it is essential to sequence the PCR products to confirm that the visible band contains the expected DNA fragments. Please provide the sequencing results for these genes.

Major Point 4

According to the Material and Method section, the authors did not use any DNAse treatment in their RT-PCR which is essential to exclude that genomic DNA contaminations might be templates in the PCR reactions. Such genomic DNA contaminations could explain the suggested intron retention which were observed in the RT PCR. The presented assays must be performed after a DNAse treatment was performed. Moreover, it must be experimentally demonstrated that no genomic DNA remains in the PCR mix by using a positive control and a negative control which should be included in the manuscript figure.

Minor Points

Abstract:

Line 20: ‚ showed higher sensitivity’ instead of ‚ showed sensitivity’. A word is missing.

Line 24: the authors write : ‘in both cultivars’. However, it was not introduced before that two cultivars were investigated … Please introduce your material to the reader, otherwise the reader is confused.

Line 31: the authors describe only a correlation therefore it is suggested to formulate that sentence less strong: ‘which suggested that the …’ rather than ‘which showed that the …’

Line 34: What do the authors mean with: ‘…There was existed interaction …’

Line 36: ‘This suggested’ instead of ‘these suggested’

Line 36ff: What do the authors mean with : ‘…splicing genes that are tolerant to B deficiency …’ How can a gene be tolerant to B deficiency?

Introduction

-Line 73: The authors write that bacteria and yeast need boron and list a reference. However, in this reference it is not stated that yeast and bacteria need boron and to the best of the current knowledge boron is not essential for these organisms.

-The RNA-seq dataset, which was evaluated in this study, was described and published already by Yuan et al. 2017. PRJNA393069. Therefore, the paragraphs 2.1, 2.2 and 2.3 in the Material and Method section are a bit misleading. The authors should refer to their previously published article within the Material and Method section.

-Table S2 describes that the authors performed a semi-quantitative PCR and in the ‘Material and Method’section they write they did a qPCR. Please correct this wrong statement.

-Line 408. ‘AS’ instead of ‘ASS’

Reviewer 2 Report

Please find some comments in the attached manuscripts
